# Fibroblast Growth Factor 23 and Outcome Prediction in Patients with Acute Myocardial Infarction

**DOI:** 10.3390/jcm11030601

**Published:** 2022-01-25

**Authors:** Anne Cornelissen, Roberta Florescu, Kinan Kneizeh, Christian Cornelissen, Elisa Liehn, Vincent Brandenburg, Alexander Schuh

**Affiliations:** 1Department of Cardiology, Angiology and Internal Intensive Medicine, University Hospital Aachen, RWTH Aachen University, 52074 Aachen, Germany; rflorescu@ukaachen.de (R.F.); kkneizeh@ukaachen.de (K.K.); eliehn@ukaachen.de (E.L.); aschuh@ukaachen.de (A.S.); 2Department of Pneumology, University Hospital Aachen, RWTH Aachen University, 52074 Aachen, Germany; ccornelissen@ukaachen.de; 3Department of Cardiology and Nephrology, Rhein-Maas Klinikum, 52146 Wuerselen, Germany; vmbrandenburg@aol.com; 4Department of Internal Medicine I, St. Katharinen Hospital Frechen, 50226 Frechen, Germany

**Keywords:** fibroblast growth factor 23, myocardial infarction, HF, GRACE score, outcome prediction

## Abstract

(1) Background: Fibroblast growth factor 23 (FGF23) is associated with mortality in patients with heart failure (HF); however, less is known about mortality associations in patients with myocardial infarction (MI). (2) Methods: FGF23 was assessed in 180 patients with acute MI, 99 of whom presented with concomitant acute HF. Patients were followed up for one year, and outcome estimates by FGF23 were compared to GRACE score estimates. (3) Results: Log-transformed serum levels of intact FGF23 (logFGF23) did not differ between MI patients with and without HF, and no difference in logFGF23 was observed between 14 MI patients who died and those who survived. However, when only MI patients with concomitant HF were considered, logFGF23 was significantly higher among non-survivors compared to that in survivors. While logFGF23 was not associated with the outcome in the entire cohort, logFGF23 was fairly predictive for one-year mortality in patients with concomitant HF (AUC 0.78; 95%CI 0.61–0.95), where it outperformed GRACE score estimates (AUC 0.70; 95%CI 0.46–0.94). (4) Conclusions: FGF23 was associated with one-year mortality only in MI patients who concomitantly presented with HF, surpassing the predictive ability of GRACE score estimates. No associations were observed in patients without HF despite similar FGF23 levels at admission. Further studies are warranted to investigate whether FGF23 is causal for dismal outcome of HF.

## 1. Introduction

Fibroblast growth factor 23 (FGF23) is a bone-marrow-derived hormone secreted by osteoblasts and osteocytes in response to increased phosphate levels, preventing phosphate reuptake in the renal proximal tubule. Furthermore, FGF23 is a potent suppressor of vitamin D hormone production, inhibiting conversion of 25-hydroxyvitamin D to its active form. Besides its role in mineral metabolism, FGF23 has been shown to have a role in cardiovascular diseases. Increased FGF23 levels have been associated not only with structural myocardial damage, including impaired left-ventricular (LV) function and adverse LV remodeling [1,2,3,4], but also with alterations in myocyte calcium handling [5], upregulation of the renin–angiotensin system [6], and promotion of vascular calcification [7]. Importantly, FGF23 may predict clinical outcome in patients with acute and established heart failure (HF) [8,9].

The exact molecular mechanisms by which FGF23 is linked to cardiovascular health are not fully understood. Recent studies suggested a direct release of FGF23 by cardiomyocytes in response to myocardial injury. Experimental myocardial infarction (MI) resulted in the upregulation of circulating intact FGF23 along with a suppression of vitamin D hormone levels in mice and in rat models [10]. Furthermore, local myocardial upregulation of FGF23 and its receptor has been shown early after MI in mice, sustained by inflammatory cytokines [11], suggesting FGF23 has a crucial role in the healing and remodeling processes after MI. On the other hand, FGF23 knockout mice develop rapid calcification and exhibit a markedly short life span [12], suggesting that any detrimental effects of FGF23 mainly occur through the myocardium.

Studies investigating the ability of FGF23 to predict outcome in patients with MI have yielded controversial results. Most importantly, it is uncertain whether FGF23 predicts outcome in patients with MI independently from the presence of HF. In patients with coronary artery disease (CAD), high FGF23 levels were predictive for the composite endpoint of acute coronary syndrome (ACS), HF, stroke, and transient ischemic attacks [13]. Elevated FGF23 was also associated with recurrent cardiovascular events, defined as a composite of cardiovascular death or hospitalization for HF, in patients recovering from ACS [14]. Of note, associations with MI were directionally consistent but of lesser magnitude [14]. Likewise, elevated FGF23 had stronger associations with chronic HF than with atherosclerotic events (including MI, stroke, and peripheral vascular disease) in a prospective cohort of CKD [15], and data from the community-based Framingham Heart Study showed that FGF23 was positively associated with all-cause mortality but not with vascular function or incident cardiovascular disease [16]. A recent study investigating FGF23 and cardiovascular outcomes in a younger (aged 45 ± 4 years) patient cohort with few comorbidities found no associations between FGF23 and incident CAD, CAD events, and mortality, whereas elevated FGF23 was independently associated with a greater risk of hospitalization for HF [17].

We sought to investigate associations between FGF23 and outcome prediction in patients with acute MI. We hypothesized that elevated FGF 23 levels are predictive for outcome in patients with acute MI and HF but are not predictive for outcome in MI patients without HF. Furthermore, we compared outcome estimates based on FGF23 to those based on the GRACE score, a well-established outcome prediction tool, in patients with acute MI with and without HF.

## 2. Materials and Methods

### 2.1. Study Population

A total of 188 patients who underwent coronary angiography for acute MI at University Hospital Aachen between August 2016 and September 2018 were included in the study. Exclusion criteria were age <18 years, pregnancy or breastfeeding, CKD requiring dialysis, heart or kidney transplantation, and primary admission to the intensive care unit or transfer to the intensive care unit for cardiogenic shock within 12 h after admission. Furthermore, patients who died within 12 h after admission were excluded from the study. In accordance with the working hours of our clinical study center, patients were screened from Mondays to Thursdays between 7 a.m. and 1 p.m.

### 2.2. Clinical Definition of Acute MI

Acute MI was defined as an elevation of cardiac troponin values with at least one value above the 99th percentile upper reference limit, indicating myocardial injury, in a clinical setting consistent with myocardial ischemia [18]. In line with the Universal Classification of Myocardial Infarction [18], each patient was further subclassified into Type I MI (spontaneous MI, related to atherosclerotic plaque rupture, ulceration, fissuring, erosion, or dissection with resulting intraluminal thrombus in at least one coronary artery) and Type II MI (MI secondary to an ischemic imbalance, including coronary artery spasm, coronary endothelial dysfunction, tachyarrhythmias, bradyarrhythmias, anemia, respiratory failure, hypotension, and severe hypertension). Patients with Type III MI (cardiac death due to MI), Type IV MI (MI related to percutaneous coronary intervention (PCI) or stent thrombosis), and Type V MI (MI related to coronary artery bypass grafting) were excluded from the study. Type I and II MI were subclassified into ST-elevation MI (STEMI) and non-ST-elevation MI (NSTEMI), both of which were defined according to the current ESC guidelines [19,20].

### 2.3. Clinical Definition of Acute HF

HF was defined in accordance with the current ESC guidelines [21]. Acute HF included patients with acute de novo HF and chronic HF with acute decompensation due to MI. Acute HF was suspected if clinical symptoms and typical signs of congestion were present in a patient (e.g., dyspnea/orthopnea, peripheral and/or pulmonary edema, elevated jugular venous pressure, hepatojugular reflux, and other signs of increased central venous pressure). NT-proBNP was assessed in all patients, and acute HF was ruled out if levels were <300 pg/mL unless echocardiography showed a reduced or mildly reduced left-ventricular ejection fraction (LVEF). All patients underwent echocardiography within 72 h after admission to determine LVEF, chamber size, regional wall motion abnormalities, right ventricular (RV) function, valve function, and markers of diastolic function. Acute HF was confirmed if patients with typical signs and symptoms either had a reduced LVEF (i.e., ≤40%; HFrEF) or mildly reduced LVEF (i.e., 41–49%; HFmrEF) or if they had a normal LVEF (i.e., ≥50%) with echocardiographic signs of LV diastolic dysfunction/raised LV filling pressures (HFpEF).

### 2.4. Laboratory Analysis and Medication Assessment

Blood samples were collected at the time of admission (“day 1”) and on the following day (“day 2”). Routine laboratory analyses were performed at our core laboratory facility at admission. In addition, serum levels of intact FGF23 were measured using a fully automated chemiluminescent assay (Liaison XL, DiaSorin S.p.A., Saluggia, Italy). Medication was assessed at discharge.

### 2.5. Study Endpoints and Follow Up

Primary endpoint:Association between FGF23 levels and survival at one year in patients with MI with and without acute HF.

Secondary endpoints:Assessment of the predictive ability of FGF23 as a biomarker for individual risk assessment and prognosis estimation.Comparison of the predictive ability of FGF23 to GRACE score estimates.All patients were followed up by phone calls to assess their survival at one year after admission.

### 2.6. Risk Calculation

We used the GRACE score [22,23] as an established risk estimation tool to estimate one-year mortality (https://www.mdcalc.com/grace-acs-risk-mortality-calculator; (22 September 2021). Variables in the equation were age, heart rate, systolic blood pressure, serum creatinine, cardiac arrest at admission, ST segment deviation on ECG, abnormal cardiac enzymes, and Killip class.

### 2.7. Statistical Analysis

Baseline characteristics were assessed by standard descriptive statistics. Continuous data are expressed as mean ± SD. Categorical data are presented as numbers and percentages. Shapiro-Wilk test was used to assess normality. Serum FGF23 levels were not normally distributed and, therefore, were log transformed (logFGF23). Outliers were identified using the ROUT method, as described elsewhere [24], based on a false discovery rate of 1%. Statistical differences between categorical variables were determined using the chi square test. Differences between continuous variables were determined with Student’s t-test or one-way ANOVA, followed by Tukey’s multiple comparisons testing for normally distributed variables, and we used Mann-Whitney and Kruskal-Wallis tests for non-normally distributed variables.

We used Spearman’s correlation to investigate the relationship between logFGF23 and GRACE score, and partial correlation was used to control for age, sex, and, eGFR. Generalized mixed model analyses were performed to assess associations between logFGF23 and survival at one year. Each model was adjusted for age, sex, and, eGFR. Furthermore, we calculated areas under the receiver operating curve (AUCs) and the corresponding 95% confidence intervals (CI) to compare the performances of the GRACE score and FGF23 for mortality prediction. All statistical analyses were performed using IBM SPSS statistics, version 28.0. Graphs were created in GraphPad Prism (Version 8.4.1, GraphPad Software, San Diego, CA, USA).

## 3. Results

### 3.1. Study Population and Patient Characteristics

A total of 188 patients admitted to the Intermediate Care Unit of University Hospital Aachen for acute MI met the initial inclusion criteria of the study. After the exclusion of one patient lost to follow up, five patients with incomplete assessments of FGF23 levels, and two outliers with extremely high FGF23 levels, the final study cohort encompassed a total of 180 patients (Figure 1).

Patient characteristics are shown in Table 1. Mean age was 65.4 ± 12.4 years (range 20–87 years), and 126 patients (70.0%) were male. Of 180 patients with acute MI, 99 patients had concomitant HF. Patients with STEMI were more likely to present with rather than without acute HF. Furthermore, patients who underwent coronary intervention of the LAD were more prone to acute HF, whereas patients with RCA lesions were less likely to present with acute HF. Patients with HF exhibited higher serum levels of markers indicating cardiomyocyte injury compared to patients without HF, consistent with a higher frequency of STEMI in patients with versus those without HF. Patients with concomitant HF were more likely to be treated with beta blockers.

The GRACE score was higher among MI patients with HF as compared to that in patients without HF, although no differences were observed in survival rates at one year. Serum levels of intact FGF23 were assessed at admission (day 1) and on the following day (day 2). LogFGF23 levels did not differ between patients with and without HF on day 1, while we observed a trend toward higher logFGF23 levels in patients without HF on day 2.

### 3.2. Associations between FGF23 Levels and Survival at One Year

In the entire cohort of 180 patients with acute MI, 14 patients did not survive the one-year follow-up period. No differences were observed in logFGF23 levels on day 1 between non-survivors and those who survived, either in the unadjusted analysis, or when accounting for the impact of age, sex, and eGFR in a generalized mixed model analysis (Table 2). However, when only MI patients with concomitant acute HF were considered, we observed significantly higher logFGF23 levels among non-survivors compared to survivors, and the difference was robust after adjusting for the above-mentioned covariates. No differences in logFGF23 levels existed among MI patients without HF.

### 3.3. GRACE Score Estimates and Correlation with FGF23 Levels

One-year mortality risk was estimated for each patient by using the GRACE score [22]. Indeed, GRACE score estimates were significantly higher among patients who did not survive the one-year follow-up period compared to that among those who survived (Table 3). However, significance for differences in GRACE score estimates between survivors and non-survivors was lost when MI patients with and without concomitant HF were considered separately.

LogFGF23 levels, regardless of whether they were assessed at admission or on day 2, were positively correlated with GRACE score estimates in the entire cohort, both in the unadjusted analysis and when age, sex, and eGFR were accounted for. While a positive correlation between logFGF23 levels and GRACE score estimates was also observed in MI patients with acute HF, and significance was lost when only patients without HF were considered (Table 4).

### 3.4. Risk Prediction Accuracy of FGF23 Levels Compared to GRACE Score Estimates

In the entire cohort, encompassing 180 patients with MI, the GRACE score correctly predicted one-year mortality in 92.2% of patients with a sensitivity of 99.4% and a specificity of 7.1% (Figure 2). While higher GRACE scores were associated with a slightly higher likelihood of death (OR 1.05; 95%CI 1.02–1.09; *p* = 0.004), the AUC value of 0.68 (95%CI 0.51–0.84) suggested poor predictive ability. Similarly, poor accuracy in risk prediction was observed with logFGF23 levels assessed at admission. In all, 92.2% patients were correctly classified by logFGF23 levels, while the AUC value of 0.64 (95%CI 0.47–0.80) suggested poor performance of FGF23 in outcome prediction at one year.

In patients with acute MI and concomitant HF, however, FGF23 levels outperformed GRACE score in risk prediction with a fair AUC value of 0.78 (95%CI 0.61–0.95) and an overall predictive accuracy of 93.9% compared with an AUC value of 0.70 (95%CI 0.46–0.94) and an overall predictive accuracy of 91.9% for GRACE score. LogFGF23 levels above 1.71 predicted death at one year with a sensitivity of 0.75 and a specificity of 0.74.

In contrast, FGF23 had no discriminative ability for prediction of survival at one year in patients without acute HF (AUC value 0.43; 95%CI 0.21–0.65) while GRACE score performed similarly poorly in predicting the one-year outcome (AUC 0.67; 95%CI 0.46–0.89).

## 4. Discussion

Numerous studies have shown associations between FGF23 and cardiovascular outcome [8,9,15,25,26,27]. While strong associations with cardiovascular morbidity and mortality have been reported in patients with HF, less is known about associations between FGF23 and outcome in patients with MI. Here, we show that systemic levels of circulating FGF23 are associated with one-year mortality only in patients with concomitant acute HF while no associations were found in MI patients without HF.

Early risk stratification can be crucial in patients with acute MI, since these patients may benefit from a closer monitoring and follow up. Several risk prediction tools have been established, for example the thrombolysis in myocardial infarction (TIMI) score [28,29], the Platelet glycoprotein IIb/IIIa in Unstable angina: Receptor Suppression Using Integrilin (PURSUIT) [30], and the GLOBAL Registry of Acute Coronary Events (GRACE) score [22,31]. Among these three risk estimation tools, the best predictive accuracy for death at one year was reported with the GRACE score [23], which was, therefore, used to estimate one-year mortality in our study. GRACE score estimates, however, had at most moderate predictive ability in our cohort. Several aspects might explain why the GRACE score performed poorly in our study. First, the GRACE score was developed in the late 1990s and early 2000s when some of the most important breakthroughs in the management of acute coronary syndromes, such as PCI with drug-eluting stents, were not yet available [32]. Accordingly, the discriminative capacity for mortality has been shown to be lower in patients with in-hospital PCI compared to that in those without PCI [33]. Moreover, a slightly lower performance of GRACE score has been reported in higher-risk cohorts such as patients with diabetes mellitus or in those with chronic kidney disease [34].

FGF23 levels were not predictive for one-year mortality in patients with acute MI when the entire cohort was considered in our analysis. In patients with concomitant HF, however, FGF23 outperformed GRACE score in risk prediction, suggesting FGF23 might potentially be useful for risk prediction in patients with acute MI and concomitant HF.

Some important aspects need to be considered when discussing our findings in the context of previous studies. Strong associations between FGF23 and cardiovascular outcome have been reported in studies using composite outcomes including HF-related endpoints and/or all-cause mortality whereas associations with atherosclerosis-related endpoints were less consistent in patient cohorts of MI [14] or coronary artery disease [25]. In community-based studies, such as ARIC [35] and REGARDS [36], higher FGF23 levels were associated with a greater risk of incident coronary heart disease events, which, however, included both acute MI and coronary heart disease–associated death, and the proportion by which acute MI contributed to this combined endpoint independently of HF is uncertain. Another recent study evaluated associations between FGF23 and different outcomes, including all-cause mortality, recurrent MI, and hospitalization for HF, in participants of the TOTAL-AMI (Tailoring of treatment in all comers with acute myocardial infarction) project [37]. While FGF23 levels were predictive for all-cause mortality and hospitalization for HF, no associations were observed with recurrent MI.

Importantly, the proportion of HF patients in previous studies may vary and is oftentimes not reported. To account for the impact of HF, it is common practice to adjust for LV ejection fraction when assessing outcomes. However, we [9] and others [38] have shown before that FGF23 does not increase linearly with declining ejection fraction. Considering that patients with HF and preserved ejection fraction had higher FGF23 levels compared with control patients and that FGF23 was also predictive for adverse cardiovascular outcome in these patients [39], the mere adjustment for ejection fraction does not fully account for the impact of HF when assessing outcomes. Therefore, we differentiated between MI patients with and without concomitant HF in our study. It is worth mentioning that the proportion of patients with acute HF was comparatively high in our study, which is supposedly owing to the fact that HF was defined according to current ESC guidelines, offering a broader definition compared to other studies of MI with concomitant HF. For example, the incidence of acute HF in patients with acute MI declined from 46% to 28% between 1996 and 2008 according to data from the SWEDEHEART (Swedish web system for enhancement and development of evidence-based care in heart disease evaluated according to recommended therapies) registry [40]. While the decline as such is supposedly owing to considerable improvements in the management of patients with MI, including early reperfusion in STEMI and improved pharmacological treatments, acute HF in SWEDEHEART was defined as “the presence of pulmonary rales, administration of intravenous diuretic agents, continuous positive airway pressure, or the use of intravenous drugs”, implying a rather severe clinical picture. Likewise, another study including 6282 patients with STEMI reported acute HF in 21.1% of patients, defined as Killip class >I [41]. For comparison, 26 out of 180 MI patients (14.4%) and 10 out of 79 STEMI patients (12.7%) presented with Killip class >I in our study. Nevertheless, we decided to follow the definition of acute HF as outlined in current ESC guidelines as they represent current clinical practice.

Cardiogenic shock represents the severest form of acute HF, and a tremendous increase in FGF23 levels has been observed in patients with acute cardiogenic shock, along with poor cardiovascular outcome [42]. By contrast, FGF23 levels in patients with uncomplicated MI were not different from those in patients with stable coronary artery disease [42] and have even been reported to decrease over the course of the first two days following an acute MI [43,44]. FGF23 levels at admission were not different between MI patients with and without HF in our cohort, although patients with concomitant HF exhibited significantly higher levels of markers indicating cardiomyocyte injury. A recent MRI study confirmed that circulating FGF23 was not associated with infarct size, LV mass, LV volumes, or LV-ejection fraction in the acute phase of first-time STEMI [44]. Interestingly, however, patients with MRI-evidence of LV remodeling 4 months after an acute STEMI had higher systemic FGF23 levels at baseline compared to those without remodeling [45]. Enhanced cardiac FGF23 mRNA and protein expression has been detected in the infarcted area following ligation of the left anterior descending (LAD) artery in a mouse model [11]. Nevertheless, LAD ligation inevitably causes HF; hence, it is still unclear whether cardiac FGF23 expression was mainly triggered by ischemia or rather by the loss of cardiac function. Our finding that FGF23 serum levels were associated with one-year mortality only in patients with concomitant HF, while no associations were observed in those without HF although serum levels were similar in both groups, raises further questions.

Faul et al. have demonstrated that intravenous and intramyocardial injection of FGF23 caused LV hypertrophy in rats [4], which increases the risk of HF and predisposes to sudden cardiac death [46]. Perhaps the effect of FGF23 is more detrimental and causes further remodeling in pre-damaged myocardium, making MI patients with concomitant HF more vulnerable to the adverse effects of FGF23. This would, however, assume a causative relationship between FGF23 and myocardial damage, which has not been proven unequivocally. In fact, LV hypertrophy itself, induced by transverse aortic constriction in a pressure overload model, has been shown to cause a profound, aldosterone-mediated increase in circulating FGF23, enhanced cardiac mRNA and protein expression of FGF23, and increased FGF23 transcription in bone [47], suggesting a more associative rather than causative relationship between FGF23 and myocardial damage. Furthermore, rare diseases with FGF23 excess such as X-linked hypophospatemic rickets (XLH), an X-linked dominant disease with FGF23 overexpression and consecutive hypophosphatemia, do not necessarily present with LV hypertrophy [48,49]. Further studies are warranted to evaluate whether or not circulating FGF23 has a causative effect on myocardial damage and what factors determine the myocardium’s susceptibility to potential detrimental effects.

Our study has some important limitations. First, the study is purely observational and does not allow for any causal interference. Second, due to the single-center design, our sample size is modest and our findings need to be validated in larger prospective cohorts. Third, because all patients were included in our study after admission for acute MI, we were unable to obtain their FGF23 levels before the event occurred. Moreover, FGF23 levels were assessed only at admission. Repeated blood sampling over the course of the convalescence period might provide additional insight into the relationship between FGF23 and adverse remodeling. Finally, we did not differentiate between cardiac and non-cardiac causes of death.

## 5. Conclusions

In patients with acute MI, serum levels of FGF23 were associated with one-year mortality only in those patients who concomitantly presented with HF, whereas no associations were observed in MI patients without HF despite similar FGF23 serum levels at admission. While FGF23 levels outperformed GRACE score estimates in outcome prediction for MI patients with HF, both FGF23 and GRACE score estimates performed equally poorly in uncomplicated MI. Further studies are warranted to investigate whether FGF23 plays a causative role in myocardial damage or whether it is a mere bystander associated with a dismal outcome.

## Figures and Tables

**Figure 1 jcm-11-00601-f001:**
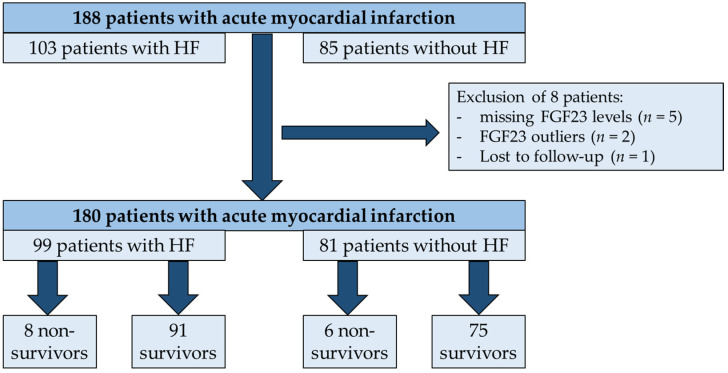
From a total of 188 patients admitted for acute MI, 8 patients were excluded. The final study cohort encompassed a total of 180 patients, 99 of whom had concomitant acute HF and 81 who did not have HF. In all, 14 patients did not survive the one-year follow-up period, and 166 patients survived.

**Figure 2 jcm-11-00601-f002:**
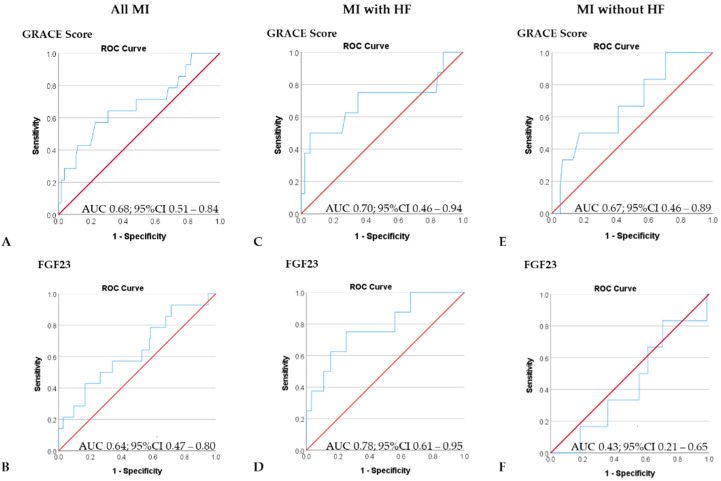
Outcome prediction of GRACE score and FGF23 levels. (**A**,**B**) In the entire cohort, encompassing 180 patients with MI, both GRACE score (**A**) and FGF23 (**B**) had limited ability to discriminate between patients who survived and patients who died during the one-year follow-up period. (**C**,**D**) While the GRACE score showed poor discriminative ability in patients with acute MI and concomitant HF (**C**), better performance in risk prediction was observed with FGF23 levels (**D**). In patients without concomitant HF, risk prediction was poor with GRACE score (**E**), while FGF23 levels did not have any ability to predict the one-year outcome (**F**).

**Table 1 jcm-11-00601-t001:** Baseline characteristics of patients with MI and HF vs. MI without HF.

	Entire Cohort (*n* = 180)	MI with HF (*n* = 99)	MI without HF (*n* = 81)	*p*-Value
Age, years	65.4 ± 12.4	66.1 ± 11.6	64.6 ± 13.2	0.40
Male sex, *n* (%)	126 (70.0%)	73 (73.7%)	53 (65.4%)	0.23
BMI, kg/m^2^	27.3 ± 4.3	27.5 ± 4.6	27.1 ± 4.4	0.52
LVEF mean ± SD (%)	47.6 ± 10.5	40.4 ± 8.4	56.4 ± 4.0	<0.001
STEMI, *n* (%)	79 (43.9%)	51 (51.5%)	28 (34.6%)	0.02
Survival, *n* (%)	166 (92.2%)	91 (91.9%)	75 (92.6%)	0.87
Coronary Intervention				
LAD, *n* (%)	82 (45.6%)	53 (53.5%)	29 (35.8%)	0.02
LCA, *n* (%)	48 (26.7%)	25 (25.5%)	23 (28.4%)	0.67
RCA, *n* (%)	68 (37.8%)	30 (30.3%)	38 (46.9%)	0.02
Medication				
Beta blockers, *n* (%)	165 (91.7%)	95 (96.0%)	70 (86.4%)	0.02
ACE inhibitors/angiotensin-II receptor blockers, *n* (%)	168 (93.3%)	93 (93.9%)	75 (92.6%)	0.72
Statins, *n* (%)	173 (96.1%)	96 (97.0%)	77 (95.1%)	0.51
Risk Factors				
Hypertension, *n* (%)	103 (57.2%)	52 (52.5%)	51 (63.0%)	0.16
Smoking, *n* (%)	97 (53.9%)	59 (59.6%)	38 (46.9%)	0.09
Diabetes, *n* (%)	45 (25.0%)	23 (23.2%)	22 (27.2%)	0.55
Hypercholesterolemia, *n* (%)	114 (63.3%)	57 (57.6%)	57 (70.4%)	0.08
Chronic kidney disease, *n* (%)	37 (20.6%)	22 (22.2%)	15 (18.5%)	0.54
Blood Parameters				
Total cholesterol, mg/dL	183.5 ± 44.6	179.4 ± 46.8	188.6 ± 41.5	0.17
LDL cholesterol, mg/dL	122.97 ± 44.94	120.37 ± 48.02	126.14 ± 40.95	0.39
HDL cholesterol, mg/dL	47.02 ± 17.04	45.89 ± 14.15	48.41 ± 20.03	0.33
Triglycerides, mg/dL	104.72 ± 67.20	100.95 ± 70.85	109.23 ± 62.70	0.41
Estimated GFR, mL/min/1.73 m^2^	89.9 ± 36.7	89.4 ± 37.3	90.4 ± 36.2	0.87
CK max., U/L	1160.9 ± 2009.9	1613.0 ± 2523.1	613.9 ± 846.3	0.001
CK-MB max., U/L	132.3 ± 153.7	171.4 ± 178.6	85.5 ± 99.7	<0.001
Troponin T max., pg/dL	2272.9 ± 3225.7	3181.5 ± 3791.2	1173.6 ± 1867.6	<0.001
LogFGF23 day 1	1.60 ± 0.24	1.60 ± 0.26	1.60 ± 0.22	0.82
LogFGF23 day 2	1.61 ± 0.29	1.57 ± 0.31	1.66 ± 0.26	0.05
GRACE Score				
Death at 12 months (%)	10.0 ± 11.4	12.0 ± 13.6	7.54 ± 7.18	0.01

**Table 2 jcm-11-00601-t002:** LogFGF23 in survivors vs. non-survivors in patients with acute MI.

	Survivors		Non-Survivors		*p*-Value	*p*-Value ^1^
	*n*	LogFGF23 Day 1	*n*	LogFGF23 Day 1		
All MI	166	1.59 ± 0.23	14	1.75 ± 0.33	0.10	0.38
MI with HF	91	1.58 ± 0.24	8	1.90 ± 0.33	0.009	0.02
MI no HF	75	1.60 ± 0.22	6	1.54 ± 0.20	0.58	0.11

^1^ Adjusted for age, sex, and eGFR.

**Table 3 jcm-11-00601-t003:** GRACE score estimates in survivors vs. non-survivors in patients with acute MI.

	Survivors		Non-Survivors		*p*-Value
	*n*	GRACE Score	*n*	GRACE Score	
All MI	166	9.09 ± 9.20	14	20.85 ± 23.87	0.03
MI with HF	91	10.56 ± 10.35	8	28.78 ± 29.36	0.07
MI no HF	75	7.32 ± 7.23	6	10.28 ± 6.33	0.33

**Table 4 jcm-11-00601-t004:** Correlation between FGF23 levels and 12-month GRACE scores in patients with acute MI.

	LogFGF23 day 1		LogFGF23 day 1 ^1^		LogFGF23 day 2		LogFGF23 day 2 ^1^	
	Correlation	*p*-Value	Correlation	*p*-Value	Correlation	*p*-Value	Correlation	*p*-Value
All MI	0.38	<0.001	0.23	0.002	0.27	0.001	0.18	0.03
MI with HF	0.45	<0.001	0.24	0.02	0.50	<0.001	0.28	0.01
MI, no HF	0.22	0.05	0.12	0.28	−0.04	0.78	−0.17	0.19

^1^ Adjusted for age, sex, and eGFR.

## Data Availability

All data are available from the corresponding author upon reasonable request.

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
