# Peer review of "Fibroblast Growth Factor 23 and Outcome Prediction in Patients with Acute Myocardial Infarction"

_jcm, 2022, doi:10.3390/jcm11030601_

Round 1
Reviewer 1 Report
The authors present a manuscript which deals with the role predictive of FGF23 in myocardial infarction. This question has been previously adressed in other manuscripts however the data till now are inconclusive.
I would strongly advice the authors to make the following changes on the manuscript to make the message more clear.
1. The number of patients presenting with concomitant heart failure in acute MI is rather high in the depicted cohort. Even more, it remains unclear how the presence of HF was definied as far as patients with cardiogenic schock were excluded. “2.3 Clinical definition of HF.” is a short paragraph – the authors should depict the HF definition in a more clear manner – dyspnea with elevated NTproBNP? NTproBNP can be elevated due to different causes – e.g. renal insufficiency and more.
2. The authors postulate that patients with STEMI were more predominant in the group with concomitant HF, however classic acute clinical decompensation with only moderately elevated enzymes but diffuse CAD is more present in elderly patients with NSTEMI due to the current literature. Please comment on this.
3. The description of diagnostics performed with the patients is short. It remains unclear, when the echo (Table 1) has been performed, at what time the medication (Table 1) was assessed (at study entry or at the discharge?), the same applies for the blood parameters (Table 1) presented.
4. The authors found that values of LogFGF23 in HF patients were higher in non-survivors. However, a multivariate analysis is missing. It remains unclear if the role of FG23 is indepenedent of e.g. age, sex, renal function. All this additional clinical information has been presented in Table 1 and should be inputed in a multivariate testing approach. May be, the role of FGF23 after MI is even less than supposed by the actually presented data/manuscript in a multivariate testing.
Author Response
Reviewer 1
The authors present a manuscript which deals with the role predictive of FGF23 in myocardial infarction. This question has been previously adressed in other manuscripts however the data till now are inconclusive.
I would strongly advice the authors to make the following changes on the manuscript to make the message more clear.
- The number of patients presenting with concomitant heart failure in acute MI is rather high in the depicted cohort. Even more, it remains unclear how the presence of HF was definied as far as patients with cardiogenic schock were excluded. “2.3 Clinical definition of HF.” is a short paragraph – the authors should depict the HF definition in a more clear manner – dyspnea with elevated NTproBNP? NTproBNP can be elevated due to different causes – e.g. renal insufficiency and more.
Answer:
We thank the reviewer for his/her comment. We agree that the clinical definition of HF warrants further explanation and added the following paragraph to section 2.3:
Lines 102-115:
“HF was defined in accordance with the current ESC guidelines [21]. Acute HF included patients with acute de novo HF and chronic HF with acute decompensation due to MI. Acute HF was suspected if clinical symptoms and typical signs of congestion were present in a patient (e.g., dyspnea/orthopnea, peripheral and/or pulmonary edema, elevated jugular venous pressure, hepatojugular reflux and other signs of increased central venous pressure). NT-proBNP was assessed in all patients, and acute HF was ruled out if levels were <300pg/mL unless echocardiography showed a reduced or mildly reduced left-ventricular ejection fraction (LVEF). All patients underwent echocardiography to determine LVEF, chamber size, regional wall motion abnormalities, right ventricular (RV) function, valve function, and markers of diastolic function. Acute HF was confirmed if patients with typical signs and symptoms either had a reduced LVEF (i.e., ≤40%; HFrEF) or mildly reduced LVEF (i.e., 41 – 49%; HFmrEF) or if they had a normal LVEF (i.e., ≥50%) with echocardiographic signs of LV diastolic dysfunction/ raised LV filling pressures (HFpEF).”
The reviewer is right with his/her observation that the number of patients with concomitant heart failure is rather high in our cohort (i.e., 99 out of 180 MI patients; 55%). We suspect the high numbers of HF patients can be explained with a broader definition of HF in our study compared to definitions used by other registries and trials. According to data from the SWEDEHEART (Swedish Web System for Enhancement and Development of Evidence-Based Care in Heart Disease Evaluated According to Recommended Therapies) Registry the incidence of acute HF in patients with MI declined from 46% to 28% between 1996 and 2008, which is supposedly owed to considerable improvements in the management of patients with MI, including early reperfusion in STEMI and improved pharmacological treatments (Desta et al. JACC Heart Fail. 2015 Mar;3(3):234-42). However, HF in SWEDEHEART was defined as “the presence of pulmonary rales, administration of intravenous diuretic agents, continuous positive airway pressure, or the use of intravenous inotropic drugs”, implying a rather severe clinical picture. Since we followed the ESC definition of acute HF, including clinical signs and symptoms, elevated NT-proBNP, and echocardiographic criteria, as outlined above, the number of patients with HF was higher compared with the SWEDEHEART data. Likewise, another study including 6,282 patients with STEMI reported acute HF in 21.1% of patients, defined as Killip class >I (Auffret et al. Int J Cardiol. 2016 Oct 15;221:433-42). For comparison, 26 out of 180 MI patients (14.4%) and 10 out of 79 STEMI patients (12.7%) presented with Killip class >I in our study. Nevertheless, we decided to follow the definition of acute HF as outlined in current ESC guidelines as they represent current clinical practice.
We added the following paragraph to the discussion section:
Lines 286-303:
“It is worth mentioning that the proportion of patients with acute HF was comparatively high in our study, which is supposedly owed to the fact that HF was defined according to current ESC guidelines, offering a broader definition compared to other studies of MI with concomitant HF. For example, the incidence of acute HF in patients with acute MI de-clined from 46% to 28% between 1996 and 2008 according to data from the SWEDE-HEART (Swedish Web System for Enhancement and Development of Evidence-Based Care in Heart Disease Evaluated According to Recommended Therapies) Registry [41]. While the decline as such is supposedly owed to considerable improvements in the man-agement of patients with MI, including early reperfusion in STEMI and improved phar-macological treatments, acute HF in SWEDEHEART was defined as “the presence of pulmonary rales, administration of intravenous diuretic agents, continuous positive air-way pressure, or the use of intravenous drugs”, implying a rather severe clinical picture. Likewise, another study including 6,282 patients with STEMI reported acute HF in 21.1% of patients, defined as Killip class >I [42]. For comparison, 26 out of 180 MI patients (14.4%) and 10 out of 79 STEMI patients (12.7%) presented with Killip class >I in our study. Nevertheless, we decided to follow the definition of acute HF as outlined in current ESC guidelines as they represent current clinical practice.”
- The authors postulate that patients with STEMI were more predominant in the group with concomitant HF, however classic acute clinical decompensation with only moderately elevated enzymes but diffuse CAD is more present in elderly patients with NSTEMI due to the current literature. Please comment on this.
Answer:
We thank the reviewer for his/her comment. In fact, the paragraph on patients with STEMI was misleading in the previous version of our manuscript. 51 out of 99 patients with HF had STEMI (51.5%), and consequently, 48 HF patients had NSTEMI (48.5%). Thus, there was no significant predominance of STEMI patients among MI patients with HF, but patients with STEMI were more likely to present with HF rather than without acute HF. We changed the sentences
“The proportion of patients with STEMI and of those with coronary intervention of the LAD was higher among patients with HF while RCA interventions were less frequent compared with patients without HF. Consistent with a higher proportion of STEMI patients and suggesting greater myocardial damage, patients with HF exhibited higher serum levels of markers indicating cardiomyocyte injury compared to patients without HF”
to (lines 171-176)
“Patients with STEMI were more likely to present with rather than without acute HF. Furthermore, patients who underwent coronary intervention of the LAD were more prone to acute HF, whereas patients with RCA lesions were less likely to present with acute HF. Patients with HF exhibited higher serum levels of markers indicating cardiomyocyte injury compared to patients without HF, consistent with a higher frequency of STEMI in patients with versus without HF.”
The second part of the reviewer’s comment relates to the fact that patients with classic acute clinical decompensation often present as NSTEMI patients with only moderately elevated enzymes but diffuse CAD, which is completely right. However, we included only those patients who underwent coronary angiography for (suspected) cardiac ischemia in an acute setting, as outlined in the paragraph “study population”. Therefore, patients with clinical decompensation and only moderately elevated enzymes would not have met the inclusion criteria unless acute angiography was performed. Thus, the proportion of patients with “classical” acute cardiac decompensation was low, explaining why we had a relatively modest proportion of NSTEMI patients among those with acute HF (48.5%).
- The description of diagnostics performed with the patients is short. It remains unclear, when the echo (Table 1) has been performed, at what time the medication (Table 1) was assessed (at study entry or at the discharge?), the same applies for the blood parameters (Table 1) presented.
Answer:
We apologize for not having described the diagnostics performed in sufficient detail. In the updated version of our manuscript, we provide further details.
Lines 109-111
“All patients underwent echocardiography within 72 hours after admission to determine LVEF, chamber size, regional wall motion abnormalities, right ventricular (RV) function, valve function, and markers of diastolic function.”
Lines 118-119
“Routine laboratory analyses were performed at our core laboratory facility at admission.”
Lines 120-121
“Medication was assessed at discharge.”
- The authors found that values of LogFGF23 in HF patients were higher in non-survivors. However, a multivariate analysis is missing. It remains unclear if the role of FG23 is indepenedent of e.g. age, sex, renal function. All this additional clinical information has been presented in Table 1 and should be inputed in a multivariate testing approach. May be, the role of FGF23 after MI is even less than supposed by the actually presented data/manuscript in a multivariate testing.
Answer:
We thank the reviewer for his/her comment. According to table 2, logFGF23 was higher in non-survivors vs. survivors only in patients with acute MI and concomitant HF (1.58 ± 0.24 vs. 1.90 ± 0.33), both in the unadjusted analysis (p = 0.009) and after adjustment for age, sex, and eGFR (p = 0.02) in multiple regression. The adjusted p-values are depicted in the last column of the table.
Submission Date
02 December 2021
Date of this review
26 Dec 2021 11:14:09
Reviewer 2 Report
Authors investigated the association between serum level of Fibroblast growth factor 23 (FGF23) and one-year mortality in patients with heart failure. Authors summarized the property of FGF23 and indicate the evidence. Authors described well about the limitation of it and this study. The reviewer has only minor concerns. Please answer this.
- Please show LDL/HDL/TG in Table 1.
- Did authors calculate the cut-off value of FGF23 with one-year mortality? Is it reasonable to use as a biomarker?
Author Response
Reviewer 2
Authors investigated the association between serum level of Fibroblast growth factor 23 (FGF23) and one-year mortality in patients with heart failure. Authors summarized the property of FGF23 and indicate the evidence. Authors described well about the limitation of it and this study. The reviewer has only minor concerns. Please answer this.
- Please show LDL/HDL/TG in Table 1.
Answer:
We thank the reviewer for his / her suggestion and added LDL-C, HDL-C, and triglyceride levels to table 1.
- Did authors calculate the cut-off value of FGF23 with one-year mortality? Is it reasonable to use as a biomarker?
Answer:
We thank the reviewer for his / her comment! In the original version of the manuscript, we did not calculate cut-off values for FGF23. In the revised version of the manuscript we show that logFGF23 levels above 1.71 predicted death at one year with a sensitivity of 0.75 and a specificity of 0.74 in our study, albeit with the limitation that larger clinical studies are warranted to verify logFGF23 as a biomarker.
The following sentence was added to the manuscript:
Lines 231-232
“LogFGF23 levels above 1.71 predicted death at one year with a sensitivity of 0.75 and a specificity of 0.74.”
Submission Date
02 December 2021
Date of this review
28 Dec 2021 06:56:42
Round 2
Reviewer 1 Report
I thank the authors for their circumstantial response.